# Home Assessment of Indoor Microbiome (HAIM) in Relation to Lower Respiratory Tract Infections among Under-Five Children in Ibadan, Nigeria: The Study Protocol

**DOI:** 10.3390/ijerph17061857

**Published:** 2020-03-13

**Authors:** Adekunle G. Fakunle, Babatunde Olusola, Nkosana Jafta, Adedayo Faneye, Dick Heederik, Lidwien A.M. Smit, Rajen N. Naidoo

**Affiliations:** 1Discipline of Occupational and Environmental Health, University of KwaZulu-Natal, 321 George Campbell Building Howard College Campus, Durban 4041, South Africa; jaftan@ukzn.ac.za; 2Department of Environmental Health Sciences, Faculty of Public Health, University of Ibadan, Ibadan 200212, Nigeria; 3Department of Virology, College of Medicine, University of Ibadan, Ibadan 200212, Nigeria; tundesji@yahoo.ca (B.O.); adedayoraji@yahoo.com (A.F.); 4Institute for Risk Assessment Sciences, Environmental Epidemiology Division (IRAS-EEPI), Utrecht University, 80177 Utrecht, The Netherlands; d.heederik@uu.nl (D.H.);

**Keywords:** home, indoor microbiome, lower respiratory tract infections, under-five children, Ibadan, study protocol

## Abstract

The association between household air pollution and lower respiratory tract infections (LRTI) among children under five years of age has been well documented; however, the extent to which the microbiome within the indoor environment contributes to this association is uncertain. The home assessment of indoor microbiome (HAIM) study seeks to assess the abundance of indoor microbiota (IM) in the homes of under-five children (U-5Cs) with and without LRTI. HAIM is a hospital- and community-based study involving 200 cases and 200 controls recruited from three children’s hospitals in Ibadan, Nigeria. Cases will be hospital-based patients with LRTI confirmed by a pediatrician, while controls will be community-based participants, matched to cases on the basis of sex, geographical location, and age (±3 months) without LRTI. The abundance of IM in houses of cases and controls will be investigated using active and passive air sampling techniques and analyzed by qualitative detection of bacterial 16SrRNA gene (V3–V4), fungal ITS1 region, and viral RNA sequencing. HAIM is expected to elucidate the relationship between exposure to IM and incidence of LRTI among U-5Cs and ultimately provide evidence base for strategic interventions to curtail the burgeoning burden of LRTI on the subcontinent.

## 1. Introduction

The microbiome of the indoor environment encompasses not only intact microorganisms such as bacteria, viruses, and molds but also microbial components and products such as mycotoxins, endotoxins, and volatile organic compounds [1]. The microorganisms within and around the human body is a portion of the microbial diversity we encounter and interact with over time, most of which takes place in the indoor environment where humans, especially children under the age of five (U-5Cs), spend more than 90% of their time [1]. Understanding the process that drives microbial communities is an important endeavor. Evidence is accumulating that both the human and environmental microbiome contribute to human health [2,3,4]. The scientific community has begun to appreciate the importance of characterizing such interactions, with increasing numbers of studies seeking to determine the biodiversity, ecology, and public health implications of the microbial congregation present in the built environment. 

Understanding the exposure–response association between the indoor airborne microbial load and lower respiratory tract infections (LRTIs) is important because this association have been found to substantially contribute to childhood morbidity and mortality [5,6] therefore posing a major challenge to the health system in sub-Saharan Africa (SSA) [7]. Each year, LRTIs cause 15% of all deaths among U-5Cs globally, and about 50% of these deaths occur in SSA [8]. Nearly 265,000 in-hospital deaths of U-5Cs took place due to LRTI globally in 2010, 99% of which were reported in developing countries [9]. Among the microbial agents associated with LRTIs, viruses play an essential role, because primary infection with viral pathogens can pre-dispose children to secondary bacterial infection [10]. Furthermore, co-infection with multiple respiratory viruses is common [11,12]. Correct and timely (viral) diagnosis can minimize the overuse of antibiotics [13]. The assessment of specific fungal [14,15] and bacterial [16] communities in homes has been demonstrated in several studies, but only a few studies have considered the home indoor microbiome (IM) [17,18,19]. In addition, there is a paucity of expertise, experience and infrastructure for suitable cutting-edge IM research in SSA. 

In order to unravel exposure-response relationships between IM and the incidence of LRTIs among U-5Cs, this study seeks to assess the abundance of IM in the homes of U-5Cs with LRTIs and compare it to homes of children without LRTIs. This manuscript describes the protocol for this study. 

### Aims, Objectives and Hypotheses 

The HAIM study seeks to test the overall hypothesis that there is a significant difference in the composition of IM in homes of U-5Cs with LRTIs compared to homes of U-5Cs without LRTIs. Based on this hypothesis, HAIM has the following four broad objectives: To test the hypothesis that the alpha-diversity measures of IM will differ in homes of U-5Cs with LRTIs compared to U-5Cs without LRTIs, our first objective is to investigate the alpha-diversity measures (richness, evenness, Shannon’s Diversity Index) of home IM among U-5Cs in Ibadan via community assessment and monitoring.We hypothesize that there are significant differences in the abundance of specific microbial taxa between cases and controls, hence our second objective is to investigate the differential abundance of specific taxa among cases and controls.To test the hypothesis that there are significant differences in the types of IM in homes of cases and controls between the wet and dry seasons, our third objective is to determine the variation in IM diversity and composition between the wet and dry seasons.To test the hypothesis that there are significant differences in the pattern of indoor environmental factors which may account for differences in LRTI incidence, occurrence, and outcome among U-5Cs, our forth objective is to describe the indoor environmental factors (such as age of homes, particulate matter (PM) concentrations, temperature, relative humidity, presence of pets, occupancy, seasonality, etc) that contribute to the variation in LRTI occurrence, incidence and outcome among U-5Cs.

## 2. Materials and Methods 

### 2.1. Study Design

A case-control study design will be used, where cases recruited will be matched to suitable controls based on age, sex, and geographical location using the inclusion and exclusion criteria described below. The matching procedure is expected to suppress the potential confounding effect of these variables on the relationship between IM and LRTI. Recruited and consented cases and controls will then be followed-up at their home within 24 h of enrollment for indoor monitoring.

### 2.2. Study Site

Participants will be selected from three health facilities in Ibadan. The three participating sites are the Otunba-Tunwase Children Emergency Clinic of the University College Hospital, Ade-Oyo Maternity Teaching Hospital; and Oni-memorial Children Hospital (as shown in Figure 1). Brief characteristics of the participating hospitals/study sites are shown below. The participating study sites receive cases of LRTI from urban, semi-urban, and rural communities. 

#### 2.2.1. Characteristics of the Three Participating Hospitals

The three hospitals are located in Ibadan. Ibadan is historically acknowledged as the largest city in sub-Saharan Africa and is one of the West African cities that are increasing by more than 100,000 inhabitants annually, a reflection of the combined effects of natural increase and net migration. The 2006 National Population Census estimated the metropolis to be inhabited by 1.34 million people while the total population of Greater Ibadan was 2.949 million [20]. The principal inhabitants of the city are the Yorubas. The city is situated at an altitude ranging from 152–213 m above sea level in a tropical rain forest. The wet season of the year runs from June through October (though occasional showers occur as early as March). In the wet season, temperature ranges from 21 °C to 31 °C, rainfall from 8.4 cm to 18.8 cm and humidity from 54% to 77%. In the dry season, temperature ranges from 20 °C to 31 °C, rainfall from 1 cm to 4.4 cm, and humidity from 43% to 83%.

#### 2.2.2. Otunba Tunwase Children Emergency Clinic (OTChew)

The Otunba Tunwase Children’s Emergency Clinic (OTChew) is located in the Department of Pediatrics of the University College Hospital (UCH), Ibadan. The University College Hospital is a 500-bed tertiary health institution and a major referral center in Southwest Nigeria. Its position makes it the most accessible tertiary and referral medical facility in Nigeria attending to a population of over 4.4 million in the South West region and beyond. The OTChew serves all socioeconomic classes of the urban population.

#### 2.2.3. Ade-Oyo Maternity Teaching Hospital

Ade-Oyo Maternity Teaching Hospital is an 80-year-old state-owned general hospital in Ibadan, Nigeria. It is highly patronized by Ibadan residents, especially those of low and middle socio-economic status. It also serves as a referral center for many primary health centers and private clinics within Ibadan and its surroundings [21].

#### 2.2.4. Oni Memorial Children’s Hospital (OMCH)

Oni Memorial Children’s Hospital (OMCH), Ibadan, is in the south west of Nigeria. It was established in 1985 as a secondary health institution and is the only state-owned children’s hospital that provides health care services exclusively for children 12 years of age and below in Oyo State.

### 2.3. Selection of Cases and Controls

Screening and recruitment will commence at the beginning of every week until four or five cases and controls each are recruited. This cycle will continue across the dry and raining seasons until a total of 200 cases and controls each are recruited. At the recruitment sites, all children arriving at the hospital will be screened, but only those younger than five will be recruited. A child will be recruited into the study only after the caregiver has consented and the purpose of the study has been explained in English or Yoruba using a consent form. Those who meet the eligibility criteria and provide us with informed consent will be included in the study. All U-5Cs confirmed by a pediatrician with LRTI and admitted to the children’s ward or emergency room, will be logged and invited to participate in the study through their caregivers. At the presenting hospitals, diagnosis of LRTI is mostly based on chest radiography. The presence of one or more of the chest radiographic features of patchy, segmental, or lobar consolidation; +/- a positive air bronchogram; and +/- pleural effusion is used to confirm the diagnosis. Controls will be primarily recruited from the same community where the cases reside. For every case of LRTI recruited from the hospitals and followed home for indoor assessment, a control will be identified, tested, and confirmed to not have LRTI and/or any of the respiratory signs and symptoms. Each control will be matched for sex, age (±3 months), and geographical location. There will be at least one control for every case recruited. Two approaches will be used in identification and recruitment of community controls. The first method will be to ask the caregiver of the recruited case to identify a neighbor whose child is of the same age group and sex as the case and then approach that caregiver with an intent to recruit her/him. After explaining the study and obtaining consent, the caregivers of such controls will be interviewed using the same instruments as cases. Second, screening of potential controls will also be performed by a pediatrician for history of fever, cough, fast breathing, and lower chest wall indrawing. Where no reported diagnosis of LRTI is identified, the caregiver will be interviewed using the same child health questionnaire used in cases. 

Cases with other systemic illnesses such as measles, symptomatic congenital heart disease, congenital malformation, or AIDS and those that present with symptoms of measles or pertussis in the preceding 10 days will be excluded from the study. Similarly, controls hospitalized for respiratory or allergic conditions or with complaints of an LRTI in the past 30 days will be excluded from the study. The HAIM protocol workflow is presented in Figure 2.

### 2.4. Caregiver Interviews

The caregivers of recruited cases and controls will be interviewed using a structured questionnaire modified from a previously validated child health questionnaire [22] to accrue basic demographic, household environment information and family smoking characteristics. The questionnaire will include information on the child/mother’s age, sex of the child, mother/father’s education, mother/father’s occupation, socioeconomic status of the family unit, household size, housing condition, cooking pattern, and parents’ smoking status. The child health questionnaire will be administered by a trained interviewer in a language more convenient to the caregiver of the child.

### 2.5. Clinical Assessment

In addition to the child health questionnaire, the health status of the child will be ascertained by a trained nurse in each of the hospitals using a clinical proforma. The proforma provides vital clinical information about the child such as breastfeeding status/duration, immunization status/vaccine received, anthropometric measurements, respiratory symptoms/signs, severity of LRTI/other diagnosis, x-ray findings, and outcome of hospital admission. This information will be obtained from both the hospital case note of the child and the caregiver.

### 2.6. Home Walkthrough Inspection 

Recruited cases and controls will be followed-up at their home within 24 h of enrollment for home survey and environmental monitoring. A validated walkthrough checklist [23] will be used by trained inspectors to collect information about household characteristics. The instrument will involve documenting real-time observations on the conditions of the house and interviewing the caregiver about information on household activities of the occupants. Housing conditions observations will include type of house, material used in the construction of roof, walls and floor, presence of doors and windows, presence of opening windows, visible mold growth, and dampness or moisture on surfaces. Activity information to be collected will included type of cooking and heating energy sources used, number of tobacco smokers in the home, use of candle or lantern for lightning, and the keeping of pets. The cooking and heating fuels used in the households will be classified as clean (electricity and liquid petroleum gas (LPG)) or dirty (kerosene and wood), and those households that used a combination of clean and dirty fuels will be classified as mixed fuel [23]. 

### 2.7. Indoor Environmental Monitoring

The indoor environmental parameters that will be assessed in the HAIM study are presented in Figure 3. The procedure for assessment of these pollutants are detailed below.

#### 2.7.1. Temperature, Relative Humidity (RH) and Particulate Matter Monitoring

Indoor environmental temperature (°C) and RH (%) will be monitored in homes of recruited cases and controls using EXTECH datalogger model 42270. The datalogger will be installed in the home over 24 h, and the readings will be retrieved using TRLog software version 4.0 (FLIR Commercial Systems Inc., Townsend West, Nashua, NH, USA). 

Particulate matter size 2.5 μm (PM_2.5_) will be sampled simultaneously as temperature and RH in the homes of cases and controls using a SKC Sidekick air sampling pump (224-52MTX Model; SKC Ltd, Blandford, Dorset, UK). Dust will be collected on 47mm Whatman membrane filters (Millipore SAS–67120–Molsheim, made in, Grand Est, France) for 24 h [24,25]. The SKC sampler operates at a flow rate of 1.7 L/min, thereby sucking a total of 2448 L of air through a filter with a diameter of 25 mm and pore size of 3.0 mm. The sampler will be mounted on a fabricated stand that is 2.0 m high above the ground level and installed in the child’s sleep/play area as identified by the caregiver. Filter papers will be equilibrated in a desiccator for 24 h and weighed before and after sampling using a microbalance. Each filter will be weighed three times to obtain a constant and accurate weight before and after sampling. Blank filters will be similarly subjected to the same conditions, and weights will be recorded. All filter handling will be done using vinyl gloves to avoid contamination. About 5% of sampled houses of cases and controls will be resampled before the end of each season to ensure the stability of the measurements.

#### 2.7.2. Active Airborne Microbiome Sampling 

The concentration of airborne bacteria and fungi will be estimated by collecting air samples using a two-stage Anderson cascade impactor (ThermoFisher Scientific, Franklin, MA, USA). Each sampler’s stage has 200 holes—1.5 mm diameter in the first stage and 0.4 mm in the second. Each stage of the sampler will be equipped with a petri dish containing agar medium (McConkey and Saboraud Dextrose agar for bacteria and fungi isolates, respectively) prepared according to the manufacturer’s specifications. All samples will be collected in the daytime at about a 1.5 m height in the room where the child sleeps/plays. The air samples will be collected at an air flow rate of 28.3 ± 2 L/min for 15 min. Samples collected will be arranged in an ice pack and transferred to the laboratory within 24 h before incubation. Cultures on McConkey Agar will be incubated using a microbiological incubator (ThermoFisher Scientific, Franklin, MA, USA) at 35 ± 2 °C for 48 h, while Saboraud Dextrose agar plates will be incubated at room temperatures for five days prior to counting. The colonies will be counted using a Quebec darkfield colony counter (Cambridge Instruments, Inc., Buffalo, NY, USA). 

#### 2.7.3. Passive Airborne Dust Sampling

Passive airborne dust samples will be collected in the homes of cases and controls using an electrostatic dust collector (EDC) (Zeeman, Utrecht, the Netherlands). The electrostatic dust collector (EDC) is an easy-to-use passive sampling device consisting of a polypropylene folder holding electrostatic cloths. Electret fibers have been shown to enhance allergen particle retention, giving the electrostatic abilities of EDCs an advantage [26]. EDCs have previously been used for sampling both low endotoxin home environments including apartments and farm homes and high endotoxin occupational environments including a companion animal hospital and a social room at a composting plant [26,27]. Each EDC sampler will contain two cloths that are rendered pyrogen-free before use by heating overnight at 200 °C and assembled following a standard protocol [28,29]. The EDCs will be placed in plastic lock sealing bags (Ziploc^®^, S.C. Johnson & Son, Racine, WI, USA) and transported to the field for sampling. Sampling will be done by opening the EDC preferably in the child’s sleeping room if different from the child’s playing area to expose the cloths to the air and allowing the collection of settling dust for 14 days at not less than 1.50 m above the floor. At the end of the sampling period, the field workers will carefully close the EDCs and return the used EDCs to the laboratory in a new plastic lock sealing bags (Ziploc^®^) and stored at −80 °C prior to qPCR analysis. Unexposed EDCs will be included and subjected to the same condition and processing which will serve as laboratory controls during deoxyribonucleic acid/ribonucleic acid (DNA/RNA) extraction and polymerase chain reaction (PCR). Similarly, 5% of the sampled houses of cases and controls will be resampled before the end of each season to ensure stability and consistency of the measurements.

### 2.8. Qualitative Detection of IM

#### 2.8.1. Bacterial Nucleic Acid Extraction, Amplification and Sequencing

To prepare samples for bacterial DNA extraction, we will cut a portion of each sampled EDC wipes and transfer them into 10 mL pyrogen free water + 0.05% Tween 20 in duplicates and vortex them rigorously for 2 min. About 5 mL of the resulting fluid will be transferred into a clean centrifuge tube and concentrated by centrifuging at 4000 rpm for 10 min. The supernatant will be discarded, and bacterial DNA will be extracted from the pellet using a commercially available bacterial DNA extraction kit following the manufacturer’s instructions. We will introduce some modifications to incorporate bead beating for mechanical disruption of the bacterial cells. We will add 500 mg of 212–300 nm glass beads (Sigma-Aldrich, Missouri, MO, USA), followed by bead beating for 60 s at maximum speed in a FastPrep (MP Biomedicals, Fishers Scientific, UK) bead beater. Reagent controls and non-exposed EDC controls will be included during the DNA extraction procedure. Extracted DNA will then be amplified with polymerase chain reaction (PCR) targeting the V3–V4 hypervariable regions of the 16S rRNA gene as described in reference [30]. Sequencing results will be quality filtered and taxonomic annotation will be performed using dada2-based QIIME2 or alike pipelines. Relative abundances of bacterial community profiles will be obtained by taxonomic classification of the obtained sequence variants (taxa) against RDB and SILVA taxonomic databases.

#### 2.8.2. Fungal Nucleic Acid Extraction, Amplification and Sequencing

To prepare samples for fungal DNA extraction, we will transfer a portion of each sampled EDC wipes into a 10 mL pyrogen free water + 0.05% Tween 20 in duplicates and vortex them rigorously for 2 min. About 5ml of the resulting fluid will be transferred into clean centrifuge tube and concentrated by centrifuging at 4000 rpm for 10 min. The supernatant will be discarded, and fungal DNA will be extracted from the pellet using a commercially available fungal DNA extraction kit following the manufacturer’s instruction. We will introduce some modifications to incorporate bead beating for mechanical disruption of the fungal cells. We will add 500 mg of 212–300 nm glass beads (Sigma-Aldrich), followed by bead beating for 60 s at maximum speed in a FastPrep (MP Biomedicals) bead beater. Reagent controls and non-exposed EDC controls will be included during the DNA extraction procedure. As previously described [31], extracted fungal DNA will then be amplified with polymerase chain reaction using primers ITS1F: CTTGGTCATTTAGAGGAAGTAA and ITS1R:GCTGCGTTCTTCATCGATG [32,33,34] targeting the internal transcribed spacer (ITS1) region of fungal ribosomal cistron. 

#### 2.8.3. Viral Nucleic Acid Extraction and Sequencing

To prepare samples for analysis of viruses, we will cut two or three squares (~8 cm × 8 cm) from each EDC wipes into smaller pieces (~2 cm^2^) and place them into a 50-mL conical tube. In order to remove viruses from the wipes, we will vigorously vortex the wipes pieces in ~20 mL of 3% beef extract and 0.05 M glycine in molecular biology grade water and then shake them for ~15 min at 200 rpm. Viral nucleic acid will be extracted using the QIAamp Viral RNA Mini Kit following the manufacturer’s protocol (Qiagen, Valencia, CA, USA). An unexposed EDC wipe and a QIAamp Viral RNA Mini Kit blank will serve as negative controls to indicate any microbial contamination in the wipes and/or extraction kit. For nucleic acid stability, immediately following RNA extraction, we will convert samples to cDNA using the iScript cDNA Synthesis Kit (Bio-Rad, Hercules, CA, USA). Prior to qPCR analysis, conventional PCR will be used to detect selected viruses using specific primers as stated in Table 1.

#### 2.8.4. Quantification of DNA/RNA 

Isolated nucleic acids will be quantified using a Nano drop workstation (ThermoFisher Scientific, Franklin, MA, USA). DNA/RNA purity is determined using 260/280 and 260/230 absorbance ratios. Pure DNA/RNA provides values of approximately 1.8, although a range of 1.6 to slightly lower than 2.0 is acceptable. Values higher than 2 suggest contamination, while significantly lower values than acceptable range suggest the presence of proteins, phenols, or other contaminants that absorb strongly at 260 nm. 

### 2.9. Quantitative Analysis Using Real-Time Polymerase Chain Reaction (qPCR)

The resulting microbial agents from the qualitative detection will be analyzed by real-time polymerase chain reaction (qPCR) using specific primers as illustrated in Table 1 to obtain the relative abundance of the individual agents. Microbial genes will be quantified using SYBR-Green-based real-time PCR with Green Master mix and UNG/low ROX (Jena Bioscience, Jena, Germany) in an iCycler iQ Real-Time PCR Detection System (ThermoFisher Scientific, Franklin, MA, USA). All procedures will be carried out according to the manufacturer’s instruction. For gene amplification, primers sequences and amplicon sizes are illustrated in Table 1. Quantification will be calculated using the standard curve Ct method. Data will be presented as log10 of fold change.

### 2.10. Statistical Analysis

The dependent variable for the HAIM study is the case/control status, which will be determined by a doctor’s diagnosis. The primary independent variable—IM diversity—will be measured in terms of microbial richness and evenness, while the covariates of interest will include variables such as household characteristics, indoor environmental parameters, daycare attendance, socio-economic status, and others demographic variables of the participants. Three common alpha-diversity measures (richness, Pielou’s evenness, and Shannon’s Diversity Index) will be calculated. Species richness (S) is defined as the number of species observed in the sample. Evenness (E) is an estimate of how equitably distributed the species abundances are. Shannon’s index (H) measures the information content of a sample and accounts for both abundance and evenness of the species present. The alpha-diversity measures account for the abundance of the individual microorganisms within the indoor environment. Correlations between the diversity measures and environmental variables would be assessed using Pearson and Spearman correlations where appropriate. This will determine whether these measures can be included jointly into regression models or whether individual models will be required for each measure. The other key exposure variable will be the indoor PM_2.5._ This will be included into the regression models both independently and in combination with the IM variables.

We will compare the mean levels of exposure (indoor microbiome diversity indices and PM_2.5_ concentrations) between cases and controls using independent sample t-test or, where appropriate, a non-parametric test and assess the association between exposure variables (indoor microbiome and PM_2.5_ concentrations) and case-control status using conditional logistic regression analyses [46] in a model at three-levels with and without adjustment for potential confounders that were not included in the matching. In the first model, association will be adjusted for household crowding, housing tenure, housing quality, presence of pets, and socioeconomic status such as family income, and in the second model, for season, indoor temperature, and indoor relative humidity. The association between case-control status and indoor microbiome composition will be further analyzed by principal coordinate analysis (PCoA) ordination of the beta-diversity (Bray–Curtis dissimilarity) metric and Permutational Multivariate Analysis of Variance. If PCoA shows a pattern of dissimilarity in the microbiome of cases and controls, a similarity percentages (SIMPER) analysis will be conducted to assess the contribution of individual taxa to the overall beta-diversity.

Figure 4 is a directed acyclic graph (DAG) detailing the causal relationships affecting the association between exposure to indoor microbiome (IM)/PM_2.5_ and lower respiratory tract infections (LRTI). The individual circle connotes an individual exposure (node) of theoretical relevance; each node is interrelated by directional arrows that represent theoretical associations based on the researchers’ assessment of priori literature and determination of biological plausibility. The association of interest, therefore, is the link represented by the green arrow connecting the exposure and outcome. Age, gender, secondhand smoke, immunization/breastfeeding status, and body mass index (BMI) (orange nodes) are theoretically causally associated with the outcome alone (ancestors of outcome). The other exposure (red node) is theoretically causally associated with both the exposure and the outcome. To adjust for confounders in the association of interest, it is necessary to block all alternate routes between the exposure and outcome (green nodes). These “adjusted variables” are then introduced into the multivariate modelling as potential confounders

### 2.11. Ethics Approval and Consent to Participate

The HAIM study was approved by the Biomedical Research Ethics Committee of the University of KwaZulu-Natal (Ref No: BE545/17), the University of Ibadan/University College Hospital Ethics Committee (Ref No.: UI/EC/17/0077), and the Oyo State Research Ethical Review committee (Ref. No.: AD13/479/462). Permission to include U-5Cs admitted in the participating hospitals was obtained from the Oyo State Ministry of Health and the authorities of the participating hospitals. Written consent to participate in the study will be obtained from the parent or legal guardian of the children. Permission for home assessment will be obtained from the parents/guardian of the child before visiting their homes. The consent form explains the purpose of the proposed research, describes the risks and discomforts involved, the expected benefits of the research, explains how confidentiality will be preserved and how the consent will be documented, informs participants that there will be no costs for investigation and lists the details of persons to be contacted about any aspect of the research. It also serves to assure the participant that should they decline to perform any or all aspects of the study requirements that they will not be penalized in any way. All information and data will be held with strict confidentiality and stored on a password-protected computer. All files will be locked in filing cabinets. To protect confidentiality, the consent forms and contact information sheets will not be stored with confidential study information but kept in a separate locked file cabinet.

## 3. Discussion

There are several innovative aspects to the HAIM study. First, the HAIM study will be the first to fully explore the diversity and composition of microbiome in homes within SSA and its association with LRTI among U-5Cs. Second, several studies from SSA have made use of active airborne sampling [47,48,49] and passive airborne sampling using petri dishes [50,51,52], but the HAIM study is the first to apply electrostatic dust collectors (EDCs) in SSA. The EDC has been employed by several studies [29,53,54] in the developed world and has been proven to be useful for assessing exposure to airborne dust and microbial constituents in home or work environments. Several studies assessing different indoor sampling approaches demonstrated that dust that settles on a standard sampler surface located above floor level is a closer representative of the actual airborne exposure [55,56] than house dust. Passive collection on EDCs has three specific advantages compared to active airborne sampling. First, particle collection onto the standardized sampler surface occurs over a discrete and known time period. Second, placing passive samplers on a sufficiently elevated surface likely captures airborne dust. Third, the EDC gives users the opportunity to capture the entire microbial composition within a particular environment, unlike the active sampling, which depends on culture media that may not support the growth of certain microorganisms. Due to these features, passive collectors of settled dust have been used in health-based studies and otherwise to assess the microbes that occupants encounter in the built environment [28,57,58,59,60]. To further buttress the effectiveness of the EDC, Normand et al., [61] in their study comparing air impaction method and the EDC, showed that a single EDC measurement is comparable to the sum of several air-impaction measurements. Therefore, EDCs provide a more precise estimation of the fungal or microbial exposition of individuals in public buildings compared with a single air impaction [61].

Third, HAIM will be the first to investigate airborne viruses in homes within SSA knowing the role of viruses in the incidence of LRTI among U-5Cs. To the best of our knowledge, indoor airborne viruses have been investigated only in the United State using SKC AirCheck pumps loaded with filter holders and a Sioutas Personal Cascade Impactor Sampler (PCIS) [62,63,64]. This environment differs substantially in climatic, meteorological and housing conditions from Ibadan where the HAIM study is based [65]. A major effort of the HAIM study is the collection of airborne viral samples from 200 homes of U-5Cs as compared to the study carried out by Prussin et al., [62] in a daycare center. In addition, the studies carried out in the United State made use of the active sampling method to sample airborne viral particles in less than 120 min, while the HAIM study intends to employ a passive sampling method using EDCs over a period of 14 days, giving ample opportunity for the collection of a good representation of specific viruses within the child’s environment. The community engagement within the HAIM study is a unique avenue to address perception and willingness of caregivers of U-5Cs to participate in home inspection and monitoring for indoor air pollutants most especially IM. The community engagement activity of the HAIM study will create awareness on the impact of exposure to IM on the respiratory health of U-5C and provide a platform to educate parents especially mothers of U-5Cs within Ibadan on the importance of home assessment for indoor airborne microbiota. 

This study may be challenged by the following: first, the recruitment of LRTI cases within the hospital setting will mean that U-5Cs with severe LRTIs may be recruited, thus excluding patients with less severe LRTIs and those who may seek alternative therapies for LRTIs outside of hospital settings. However, the community survey strategy is designed to eliminate this bias. Second, recruiting matched controls from the same catchment area as the cases could introduce some selection bias due to the likelihood of the ideal controls not consenting to participate in the study. The study will make an effort to mitigate such an occurrence by making sure that the caregivers of the controls are well informed of the benefits of participating in the study. Third, responses from parents/caregivers of a child with severe conditions may be influenced by the state of the child, thereby introducing some response bias. The study will ensure that such situations are alleviated by making sure the administration of the tools is carried out in a conducive and comfortable atmosphere.

## 4. Conclusions

The HAIM study will elucidate the relationship between exposure to IM and incidence of LRTI among U-5Cs and ultimately provide evidence base for strategic interventions to curtail the burgeoning burden of LRTI on the subcontinent.

## Figures and Tables

**Figure 1 ijerph-17-01857-f001:**
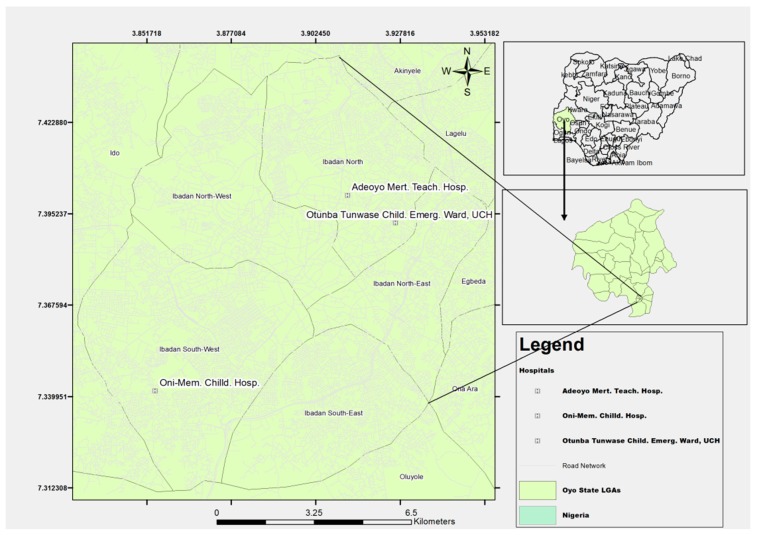
Map of Ibadan showing the location of participating hospitals. The figure was originally created for the purpose of the study.

**Figure 2 ijerph-17-01857-f002:**
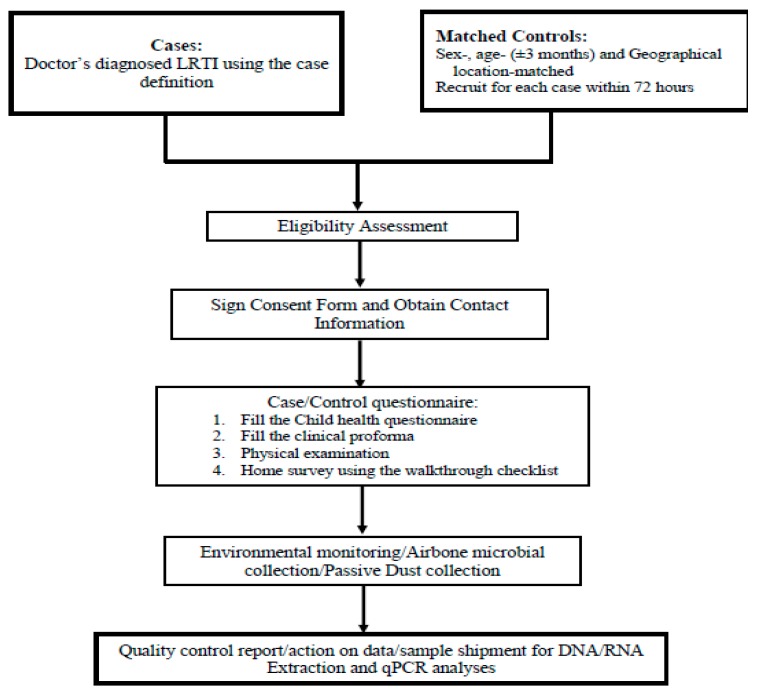
Home assessment of indoor microbiome (HAIM) protocol workflow for cases and controls. LRTI, Lower Respiratory Tract Infections; DNA, Deoxyribonucleic acid; RNA, Ribonucleic acid; qPCR, quantitative polymerase chain reaction.

**Figure 3 ijerph-17-01857-f003:**
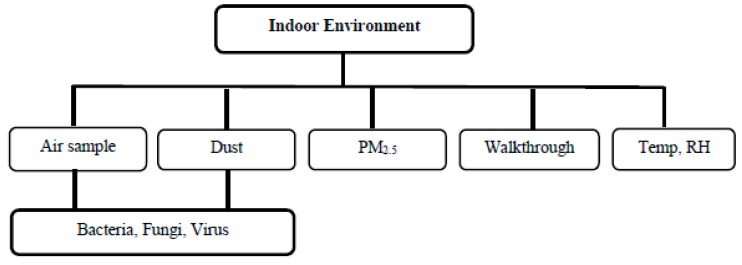
Diagram showing indoor air pollutants that will be assessed in the HAIM project. Temp, Temperature; RH, Relative Humidity.

**Figure 4 ijerph-17-01857-f004:**
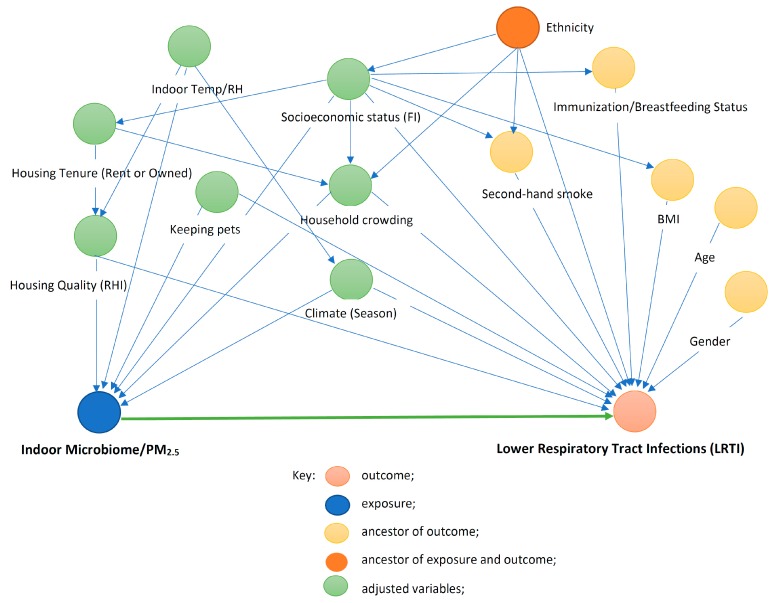
Directed acyclic graph (DAG) demonstrating causal relationships affecting the association between exposure to indoor microbiome (IM)/PM_2.5_ and lower respiratory tract infections (LRTI). IM, indoor microbiome; PM_2.5,_ particulate matter size 2.5 µm; RHI, respiratory hazard index; indoor Temp, indoor temperature; indoor RH, indoor relative humidity.

**Table 1 ijerph-17-01857-t001:** Primers for real-time polymerase chain reaction (PCR) analysis.

S/N	Organism	Name of Primer	Sequence	Size	Source
1	*Streptococcus pneumoniae*	CpsA-F	GCAGTACAGCAGTTTGTTGGACTGACC	160	[35]
		CpsA-R	GAATATTTTCATTATCAGTCCCAGTC		
2	*Streptococcus pyogenes*	DnaseB-F	TGATTCCAAGAGCTGTCGTG	140	[36]
		DnaseB-R	TGGTGTAGCCATTAGCTGTGTT		
3	*Staphylococus aureus*	THERM-F	ATGCAAAGAAAATTGAAGTCGA	233	[37]
		THERM-R	GCGTTGTCTTCGCTCCAAAT		
4	*Haemophillus influenza*	Hia-F	GCAACCATCTTACAACTTAGCGAATAC	83	[38]
		Hia-R	GGTCTGCGGTGTCCTGTGTT		
5	*Klebsiella pneumoniae*	Khe-F	GATGAAACGACCTGATTGCATTC	77	[39]
		Khe-R	CCGGGCTGTCGGGATAAG		
6	*Moraxella catarrhalis*	16SRNA-F	TTGGCTTGTGCTAAAATATC	140	[40]
		16SRNA-R	GTCATCGCTATCATTCACCT		
7	*Aspergillus/Peniclilium* spp	AspPenF1	GTCCGAGCGTCATTTCTG	228	[41]
		AspPenF2	TCCGAGCGTCATTGCTG		
8	*Fusarium spp*	Fus1	TCCATWGCGTAGTAGTAAAACCC	132	[41]
		Fus2	TCCATYGCGTAGTAGCTAACACC		
9	*Cladosporium spp*	Clado-SYBRG-PF	TACTCCAATG GTTCTAATATTTTCCTCTC	87	[42]
		Clado-SYBRG-PR	GGGTACCTAGACAGTATTTCTAGCCT		
10	RSV n gene	RSVF	GGCAAAT ATGGAAACATACGTGAA	84	[43]
		RSVR	TCTTTTTCTAGGACATTGTAYTGAACAG		
11	Para influenza 1 ngene	PIV1NF	TCTGGCGGAGGAGCAATTATACCTGG	84	[44]
		PIV1NR	ATCTGCATCATCTGTCACACTCGGGC		
12	Parainfluenza 2	PIV2NF	GATGACACTCCAGTACCTCTTG	197	[44]
		PIV2NR	GATTACTCATAGCTGCAGAAGG		
13	Human metapneumovirus	HMPVNF	GTGATGCACTCAAGAGATACCC	199	[45]
		HMPVNR	CATTGTTTGACCGGCCCCATAA		
14	Influenza a matrix	INFAF	AGGYWCTYATGGARTGGCTAAAG	105	[44]
		INFAR	GCAGTCCYCGCTCASTGGGC

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
