# Peer review of "Home Assessment of Indoor Microbiome (HAIM) in Relation to Lower Respiratory Tract Infections among Under-Five Children in Ibadan, Nigeria: The Study Protocol"

_ijerph, 2020, doi:10.3390/ijerph17061857_

Round 1
Reviewer 1 Report
This is a well written protocol paper.
My only question/ comments relates to the stability of these measurements over time. I think the authors should consider a resampling of a small portion of cases and controls at some interval ( say 6 months) after the original sampling.
Author Response
We are indeed indebted to the reviewer for spending quality time and effort in constructively reviewing our manuscript and offering useful suggestions to improve our manuscript. We have taken into account all points raised by the reviewers in revising our manuscript. The revisions are presented below in a tabular form itemizing each of the reviewer’s comment and our reply in addition to indicating the revision(s) made and the sections of the manuscript where revisions were made.
S/N |
Reviewers’ Comments |
Authors’ Reply |
Corrections made |
1 |
This is a well written protocol paper |
Thank you so much for this encouraging statement. |
|
2 |
My only question/comments relates to the stability of these measurements over time. I think the authors should consider a resampling of a small portion of cases and controls at some interval (say 6 months) after the original sampling. |
We agree absolutely. Therefore, about 5% of sampled houses of cases and controls would be resampled before the end of each season to ensure stability and consistency of the measurements.
|
Page 6; line 211-212 Page 7; line 244-246 |
Reviewer 2 Report
In this manuscript, authors described a protocol to study the relationship between indoor microbiome and lower respiratory tract infections for children under 5 years old. Authors provided very detailed description about their hypothesis, sampling and experiment design, and statistical analysis procedures. Authors also thoroughly discussed the novelty and potential drawbacks of the work. I think the protocol suggested by authors is feasible and the data collected using this protocol is adequate to test authors' hypothesis. And the proposed work is going to add new and valuable contribution to the current literature body. Topic of the manuscript also fits the scope of the journal. Therefore, I would like to recommend the publication of this manuscript.
Author Response
We are undeniably grateful to the reviewer for spending quality time and effort to review our manuscript and we appreciate the recommendation of the reviewer to publish the manuscript.
Reviewer 3 Report
The manuscript describes a study protocol to investigate the association between indoor microbial characteristics and lower respiratory tract infections (LRTI) among children under-five years of age. It is relatively well-written and worth publishing. However, authors should address the following points to make the manuscript further improved.
The authors aim to evaluate the “exposure-response relationships” between indoor microbiota and LRTI in this study. However, the manuscript is missing the details in how they assess the exposure-response relationships. More details should be presented in the method section, especially in the statistical analysis part.
What is the waiting time for medical services in Nigeria? Could access to medical services impact on recruitment of LRTI cases? Authors use the term “incidence” of LRTI, however the term “incidence” is not appropriate when the waiting time for medical services is too long. Besides, the authors should provide information about specificity/sensitivity of LRTI diagnosis in Nigeria.
The authors should provide more details about the potential confounders. Using a directed acyclic graph (DAG) would be helpful to illustrate the relationships among indoor microbiota, LRTI and potential confounders.
The authors should elaborate more clearly what kind of selection biases existed and how they were mitigated in the study.
2.1 Aims, Objectives and Hypotheses: Authors should move this section to introduction section.
Author Response
We appreciate the reviewer for spending quality time and effort in reviewing our manuscript and proposing valuable suggestions to advance our manuscript. We have taken into account all points raised by the reviewer in revising our manuscript. The revisions are presented below in a tabular form itemizing each of reviewer’s comment and our reply in addition to indicating the revision(s) made and the sections of the manuscript where revisions were made.
S/N |
Reviewers’ Comments |
Authors’ Reply |
Corrections made |
1 |
The authors aim to evaluate the “exposure-response relationships” between indoor microbiota and LRTI in this study. However, the manuscript is missing the details in how they assess the exposure-response relationships. More details should be presented in the method section, especially in the statistical analysis part. |
In order to assess the exposure-response relationship between indoor microbiome and LRTI, we will compare the mean levels of exposure (indoor microbiome diversity indices and PM2.5 concentrations) between cases and controls using independent sample t-test or, where appropriate, a non-parametric test and assess the association between exposure variables (indoor microbiome and PM2.5 concentrations) and case-control status using conditional logistic regression analyses in a model with three-levels. In the first model, association will be adjusted for household crowding, housing tenure, housing quality, presence of pets and family income, and in the second model for season, indoor temperature and indoor relative humidity.
The association between case-control status and the indoor microbiome composition will be further analyzed by principal coordinate analysis (PCoA) ordination of the beta-diversity (Bray-Curtis dissimilarity) metric and Permutational Multivariate Analysis of Variance. If PCoA shows a pattern of dissimilarity in the microbiome of cases and controls, a SIMilarity PERcentages (SIMPER) analysis will be conducted to assess the contribution of individual taxa to the overall beta-diversity.
Sequencing results will be quality filtered and taxonomic annotation will be performed using dada2-based QIIME2 or alike pipelines. Relative abundances of bacterial community profiles will be obtained by taxonomic classification of the obtained sequence variants (taxa) against RDB and SILVA taxonomic databases.
|
Page 1; line 332 – 344
page 7; line 262 - 265 |
2 |
What is the waiting time for medical services in Nigeria? Could access to medical services impact on recruitment of LRTI cases? Authors use the term “incidence” of LRTI, however the term “incidence” is not appropriate when the waiting time for medical services is too long. Besides, the authors should provide information about specificity/sensitivity of LRTI diagnosis in Nigeria. |
The hospitals run a special clinic for children under-five years of age. A case of LRTI or other conditions is directly channeled through the children's emergency clinic where they receive instant attention and treatment by a Paediatrician. Therefore, the waiting time is negligible and has no impact on recruitment. The diagnosis of LRTI is mostly based on chest radiography. The presence of one or more of the chest radiographic features of patchy, segmental or lobar consolidation, +/- a positive air bronchogram and +/- pleural effusion is used to confirm the diagnosis (Abdulkarim et al., 2013) |
Page 4; line 141 - 144 |
3 |
The authors should provide more details about the potential confounders. Using a directed acyclic graph (DAG) would be helpful to illustrate the relationships among indoor microbiota, LRTI and potential confounders. |
The DAG demonstrating causal relationships and potential biasing pathways affecting the association between exposure to indoor microbiome (IM)/PM2.5 and lower respiratory tract infections (LRTI) have been included |
Page 2; line 357 - 359 |
4 |
The authors should elaborate more clearly what kind of selection biases existed and how they were mitigated in the study. |
Recruiting matchable controls from the same catchment area as the cases could introduce some selection bias due to the likelihood of the ideal controls not accepting to participate in the study. Therefore, the study will make effort to mitigate such occurrence by making sure that the caregivers of the controls are well informed of the benefits of participating in the study. |
Page 3; line 421 - 423 |
5 |
2.1 Aims, Objectives and Hypotheses: Authors should move this section to introduction section. |
The aims, objectives, and hypothesis have been moved to the introduction section |
Page 2; line 65 - 84 |